# Comparative Transcriptomic Analyses Reveal Key Pathways in Response to Cold Stress at the Germination Stage of Quinoa (*Chenopodium quinoa* Willd.) Seeds

**DOI:** 10.3390/plants14081212

**Published:** 2025-04-15

**Authors:** Rao Fu, Xiaoyan Liang, Jiajia Li, Yanjing Song, Kuihua Yi, Wenjing Nie, Lan Ma, Junlin Li, Meng Li, Xiangyu Wang, Haiyang Zhang, Hongxia Zhang

**Affiliations:** 1Yantai Key Laboratory for Evaluation and Utilization of Silkworm Functional Substances, Yantai Engineering Research Center for Plant Stem Cell Targeted Breeding, Shandong Institute of Sericulture, 5 Qingdao Avenue, Yantai 265503, China; frao2017@163.com (R.F.); liangxiaoyan1001@163.com (X.L.); jjli7525@163.com (J.L.); yjsong1214@163.com (Y.S.); kuihuayi1016@163.com (K.Y.); cottage1990@163.com (W.N.); cysmalan@shandong.cn (L.M.); lijunlin517@163.com (J.L.); lim0320@126.com (M.L.); cyswangxiangyu@shandong.cn (X.W.); 2State Key Laboratory of Nutrient Use and Management, Institute of Agricultural, Resources and Environment, Shandong Academy of Agricultural Sciences, 23788 Gongye North Road, Jinan 250100, China; 3National Center of Technology Innovation for Comprehensive Utilization of Saline-Alkali Land, Dongying 257345, China; 4The Engineering Research Institute of Agriculture and Forestry, Ludong University, 186 Hongqizhong Road, Yantai 264025, China

**Keywords:** cold stress, quinoa, transcriptomic analysis, MAPK signaling, reactive oxygen species

## Abstract

Quinoa (*Chenopodium quinoa* Willd.) has been widely grown as a cash crop. However, the molecular mechanism by which it responds to cold stress at the seed germination stage is still largely unknown. In this study, we performed a comparative transcriptomic analysis between the cold-tolerant cultivar XCq and cold-sensitive cultivar QCq in response to cold stress. A total number of 4552 and 4845 differentially expressed genes (*DEGs*) were identified in XCq and QCq upon the treatment of cold stress, respectively. Kyoto Encyclopedia of Genes and Genomes (KEGG) pathway analysis demonstrated that the mitogen-activated protein kinase (MAPK) signaling pathway was identified only among the up-regulated DEGs in XCq.The expression of DEGs, which encoding transcription factors, such as AP2/ERF, bHLH, bZIP, MYB, ICEs, and CORs related to cold response, were higher in XCq than in QCq in response to cold stress. Weighted gene co-expression network analysis (WGCNA) showed that DEGs clustered in the co-expression modules positively correlated with the factors of quinoa variety and temperature were significantly enriched in the oxidative phosphorylation metabolic pathway. Further physiochemical analyses showed that the activities of superoxide dismutase and peroxidase as well as the contents of soluble protein and sugar, were significantly higher in XCq than in QCq. In summary, MAPK signaling and oxidative metabolism were the key pathways in quinoa upon cold stress. Our findings revealed that the enhanced activities of antioxidant enzymes alleviate the lipid peroxidation of membranes and promote the accumulation of osmotic adjustment substances, thereby enabling seeds to better resist oxidative damage under cold stress.

## 1. Introduction

Among the key environmental factors, cold stress exerts a significant influence on plant growth, biomass production, and geographical distribution [1,2,3]. It alters the structure and stability of the membrane and protein and causes chloroplast photo-oxidative damage by promoting the production of reactive oxygen species in plant cells [4,5]. In higher plants, seed germination, which could be affected by a series of environmental factors including cold stress, is the first stage in the life cycle. Under optimal water and oxygen conditions, temperature is one of the key external factors that affected seed germination [6]. When cold stress is encountered, seed germination can be delayed, reduced, or restrained, leading to uneven emergence and decreased yield and quality [7,8].

As sessile organisms, plants have evolved sophisticated mechanisms to acclimate to adverse environmental conditions, including cold stress. Through antioxidant biosynthesis, intracellular osmotic protection substance accumulation, and physical structure adaptation, their tolerance to cold stress can be significantly improved [6,9]. When exposed to stress conditions, reactive oxygen species (ROS) generation and phytohormone imbalance are the primary responses of plant cells to adverse environmental stress. To date, physiological, transcriptomic, and metabolomic studies on seed germination in response to cold stress have been conducted in various plants [6,10,11]. In maize, the activity of antioxidant enzymes, such as superoxide dismutase (SOD), peroxidase (POD), and catalase (CAT), and the contents of proline and soluble sugar were significantly increased, leading to the scavenging of ROS and improving the resistance to cold stress [7]. The contents of antioxidant enzymes, proline, and soluble sugar and the expression of genes involved in plant hormones and the mitogen activated protein kinase (MAPK) signaling pathway were significantly higher in the cold-tolerant cultivar [6]. In rice, guvermectin improved seed germination under low temperature conditions, by regulating the contents of gibberellin (GA), abscisic acid (ABA), soluble sugar, and protein, enhancing the activities of antioxidant enzymes and activating gibberellin responsive transcription factors [10]. In cotton, the transcripts of differentially expressed genes (DEGs) were significantly enriched in energy metabolic pathways such as glycolysis/gluconeogenesis and the glyoxylate cycle during seed germination [11]. In addition, the number of DEGs encoding antioxidants and antioxidase in the cold-tolerant cultivar was dramatically greater than that in the cold-susceptible cultivar [11].

A vast array of cold-responsive TFs, such as NAC, WRKY, AP2/ERF, MYB, and bHLH, have been identified in various model plants and crops. In *Arabidopsis*, DREB/C-repeat factor (CBF), a member of the AP2/ERF TF family, plays a prominent role in cold acclimation [12]. Under cold stress conditions, the expression of *CBF* genes is induced, which subsequently regulates the expression of downstream cold-related genes [13]. Furthermore, the MAPK signaling pathway functions as an important mechanism in plant response to cold stress [14,15]. The cold-activated MPK3 and MPK6 negatively regulate cold response by phosphorylating the MYC-like bHLH TF family member ICE1, which subsequently regulates the expression of CBF genes, whereas MPK4 positively regulates cold response by constitutively suppressing the activity of MPK3 and MPK6 [15]. In maize, the transcription levels of genes in the MAPK signaling pathway were significantly higher in the cold-resistant variety than those in the cold-sensitive variety [6].

Quinoa, originally from the Andean regions, has earned special attention due to its nutritional and health benefits, as well as its high resistance to abiotic stress [16,17]. The adaptability of quinoa to various environmental conditions is largely attributed to its broad genetic variability [18]. Quinoa shows multiple adaptations, from physiological to morphological levels, which serve a range of responses to stress conditions. The synthesis of antioxidant enzymes and the accumulation of the cell osmotic potential regulator are the major mechanisms in response to drought stress in quinoa [19,20,21]. The effects of temperature fluctuation on quinoa seed germination have been investigated in previous studies [22,23]. Based on the studies in different quinoa varieties, a positive correlation between temperature and seed germination, with an optimal germination temperature of 18–35 °C and the lowest germination temperature of 1 to 3 °C, has been observed [17,22,24]. Under cold stress conditions, quinoa shows different mechanisms to prevent it from being damaged [25]. One of the main mechanisms is the avoidance of ice formation, which could be ensured by the high soluble sugar content in plant cells [26]. The existence of soluble sugar, such as sucrose, prohibits ice formation in cells and improves the tolerance to freezing temperatures [26].

To unveil the molecular mechanism of plants in response to cold stress, RNA-sequencing (RNA-seq) and transcriptomic analyses have been conducted in various plants [6,11,27,28,29]. The relationship between co-expressed modules and phenotypes was investigated in sugarcane seedlings [30]. Comparative transcriptomic and metabolomic analysis in the seedlings of cold-sensitive and cold-tolerant cultivars indicated that, by accumulating more soluble sugar, energy substrate, and α-linolenic acid, normal metabolism is maintained to deal with the damage caused by cold stress [29]. Recently, a number of core genes associated with low-temperature stress were also identified in quinoa seedlings [31]. We conducted transcriptome sequencing and weighted gene co-expression network analysis (WGCNA) of gene expression patterns and physiological indicators in response to cold stress in germinating quinoa seeds. We showed that genes involved in the MAPK signaling and oxidative phosphorylation metabolic pathways play a key role in the response to cold stress at the germination stage of quinoa seeds. Our results will provide fundamental information for future studies on the functions of potential candidate genes associated with cold tolerance in plants.

## 2. Results

### 2.1. Different Germination Rates Are Observed in XCq and QCq

To evaluate the resistance of quinoa cultivars XCq and QCq to cold stress, seeds were germinated for different time periods under normal (20 °C) and cold stress (4 °C) conditions. The effects of cold stress on seed germination were compared. Under normal conditions, a faster germination was observed in XCq. After 12 h, both XCq and QCq started to germinate, with no significant difference in germination rates observed. After 15 h, a higher germination rate was observed in XCq. The higher germination rate in XCq remained until 36 h; then, no significant difference in germination rates was observed between the two cultivars after 48 h (Figure 1A). Upon cold stress treatment, seed germination in both cultivars was restrained. Seeds of XCq started to germinate after 60 h, but seeds of QCq remained un-germinated until 120 h. During this period, the germination rate of XCq was significantly higher than that of QCq. Even after 144 h, more than 97.33% of XCq seeds, but only 6% of QCq seeds, germinated (Figure 1A,B). These observations suggest that XCq was more tolerant to cold stress than QCq at the germination stage.

### 2.2. A Great Number of Differentially Expressed Genes Are Identified in XCq and QCq

To understand the reasons for the different cold tolerances of XCq and QCq, we performed RNA-Seq analysis. The transcriptional differences of genes in the embryos of XCq and QCq seeds germinated under normal and cold stress conditions for 36 and 76 h were examined. A total number of 4552 and 4845 DEGs, with 2494 of them found in both varieties, were identified in XCq and QCq, respectively. Among these DEGs, 1644 and 2908 DEGs in XCq, and 1438 and 3407 DEGs in QCq, were up- and down-regulated, respectively (Figure 2A–D). In general, more genes were differentially expressed in QCq than in XCq. Among these DEGs, more DEGs were down-regulated in QCq and more DEGs were up-regulated in XCq.

### 2.3. DEGs in XCq and QCq Are Enriched in Different Biological Processes and Pathways

We further performed Gene Ontology (GO) enrichment analyses for the identified DEGs between the seed embryos of XCq and QCq germinated at 20 °C and 4 °C. The largest number of up-regulated DEGs in the two quinoa cultivars was enriched in the biological process of “response to temperature stimulus” (GO:0009266). DEGs enriched in the four biological processes “seed dormancy process (GO:0010162)”, “seed maturation (GO:0010431)”, “import across plasma membrane (GO:0098739)”, and “regulation of seed dormancy process (GO:2000033)” were only observed in XCq. In addition, down-regulated DEGs were mainly enriched in the biological processes related to plant growth and development, such as “regulation of growth (GO:0040008)”, “regulation of meristem growth (GO:0010075)”, and “regulation of developmental growth (GO:0048638)” in XCq (Figure 3A,B). KEGG pathway analysis of the common DEGs in XCq and QCq revealed that the top three significantly enriched pathways of up-regulated DEGs were “Photosynthesis proteins”, “Photosynthesis”, and “Oxidative phosphorylation”, while the top three significantly enriched pathways of down-regulated DEGs were “Plant hormone signal transduction”, “Cytochrome P450”, and “Glycosyltransferases” in both varieties under cold stress conditions (Figure 3C,D). As important pathways implicated in plant response to biotic and abiotic stress, DEGs enriched in the MAPK signaling pathway were up-regulated, whereas DEGs involved in the starch and sucrose pathway were down-regulated, in XCq only, suggesting that both MAPKs and sugars could play a key role in response to cold stress in quinoa.

MAPKs function in plant response to low temperature by regulating the ICE1-CBF-COR cascade. In crop plants, the ICE-CBF-COR cascade works as a universal pathway related to cold stress tolerance. Based on the sequence similarity with the *Arabidopsis* proteins and the presence of DNA-binding domains, genes related to ICE-CBF-COR pathway in quinoa were detected. The expression levels of these genes involved in the ICE-CBF-COR pathway under normal and cold stress conditions were analyzed to determine their association with cold acclimation. A total of 58 DEGs possibly involved in the ICE-CBF-COR pathway, including 19 MAPKs, 10 ICEs, 9 MYBs, 6 CBFs, and 14 COR genes, were identified (Appendix A).

### 2.4. Transcription Factor Encoding Genes Are Differentially Expressed in XCq and QCq

We further analyzed the transcription factor (TF) genes in the identified DEGs. Among the 1644 and 1438 up-regulated DEGs in XCq and QCq, 89 and 45 TF genes, including 16 AP2/ERF (11 in XCq and 5 in QCq), 15 bHLH (7 in XCq and 8 in QCq), 11 bZIP (8 in XCq and 3 in QCq), 11 HSF (7 in XCq and 4 in QCq), 8 MYB (6 in XCq and 2 in QCq), 5 NAC (4 in XCq and 1 in QCq), and 3 WRKY (2 in XCq and 1 in QCq), were differentially expressed (Figure 4A). However, among the 2908 and 3407 down-regulated DEGs in XCq and QCq, 137 and 198 TF genes, including 37 AP2/ERF (12 in XCq and 25 in QCq), 38 bHLH (18 in XCq and 20 in QCq), 14 bZIP (8 in XCq and 6 in QCq), 4 HSF (0 in XCq and 4 in QCq), 18 MYB (7 in XCq and 11 in QCq), 23 NAC (12 in XCq and 11 in QCq) and 33 WRKY (8 in XCq and 25 in QCq), were differentially expressed (Figure 4A). The expression of most AP2/ERF, bHLH, bZIP, NAC, MYB, and WRKY encoding genes was higher, whereas the expression of all HD-ZIP encoding genes was lower, in XCq than in QCq in response to cold stress (Figure 4B; Appendix A).

### 2.5. Expression of DEGs Related to Cold Stress Are Closely Correlated with the Quinoa Variety and Temperature

To generate the co-expression modules of the DEGs identified in XCq and QCq, we conducted WGCNA using the hierarchical clustering method. Nine valid co-expression modules were obtained (Figure 5A). DEGs clustered in the turquoise and brown modules, with a correlation value of 0.95 and 0.85 (*p* < 0.01), respectively, were positively correlated with the temperature factor, whereas DEGs clustered in the pink module, with a correlation value of 0.81 (*p* < 0.05), were positively correlated with the quinoa variety factor. In contrast, DEGs clustered in the yellow and green yellow module, with a correlation value of 0.76 and 0.73 (*p* < 0.01), respectively, were negatively correlated with the quinoa variety factor. Gene expression analyses showed that the expression of up-regulated DEGs clustered in the nine modules was higher in XCq than that in QCq (Figure 5B).

Based on the observation that DEGs clustered in the turquoise and pink modules showed the greatest correlation with the quinoa variety and temperature factors, respectively, we further performed KEGG enrichment analysis (Figure 5C). We found that DEGs clustered in the pink module were mainly enriched in pathways related to “Chemical carcinogenesis-reactive oxygen species” (ko05208), “Oxidative phosphorylation” (ko00190), “Plant hormone signal transduction” (ko04075), and “Thermogenesis” (ko04714). DEGs clustered in the turquoise module were mainly enriched in “Photosynthesis proteins” (ko00194), “Photosynthesis” (ko00195), “Oxidative phosphorylation” (ko00190), and “Steroid biosynthesis” (ko00100).

### 2.6. Antioxidant Resistance Is Closely Correlated with Quinoa Variety and Cold Stress

Plants have developed sophisticated mechanisms, including the activation of antioxidant systems, to counteract ROS production and alleviate oxidative damage caused by environmental stress. We examined the activities of antioxidant enzymes in the seed embryos at germination stages of 36 h and 76 h. Under normal germination conditions (20 °C), no significant difference in the activities of SOD, POD, and CAT or the content of soluble protein were observed between XCq and QCq. However, under cold stress conditions (4  °C), higher activities of SOD and POD, accompanied by a lower content of MDA and higher contents of soluble protein and sugar, were observed in XCq compared to QCq (Figure 6A–F). No significant difference was found in the content of proline between XCq and QCq under both normal and cold stress conditions (Figure 6G). We also conducted Pearson correlation analysis and found that the activities of SOD, POD, and CAT as well as the content of soluble protein were positively, whereas the content of MDA was negatively, correlated with cold stress treatment (Figure 6H).

### 2.7. RNA-Seq Data Are Validated with qRT-PCR Assays

To confirm the reliability of RNA-seq data, qRT-PCR analysis was carried out. The transcription levels of 12 cold tolerance-associated genes, including three genes (AUR62034546, AUR62018879, AUR62024720) involved in the ICE-CBF-COR pathway, two genes (AUR62001348, AUR62030413) involved in the biosynthesis of antioxidant enzymes, one gene (AUR62008699) involved in glucose metabolism, and fiveTF genes (AUR62038742, AUR62024174, AUR62004404, AUR62001531, AUR62035906) involved in low-temperature response, were investigated in the two quinoa varieties. Consistent expression patterns with those generated from RNA-seq data were obtained (Figure 7). Therefore, the RNA-seq data are reliable for the analyses conducted in this study.

## 3. Discussion

Cold stress is considered to be a major factor limiting the regular growth and development of plants. As a typical Andean crop, the germination and seedling establishment of quinoa are severely affected by low temperature for the early spring sowing and planting in high altitude in cold regions [16,17,29]. Previous physiological mechanisms and transcriptome study demonstrated that α-linolenic acid metabolism was up-regulated, whereas the accumulation of energy substrates was reduced in quinoa seedlings upon low-temperature treatment [29]. To dissect the molecular mechanism by which quinoa responds to cold stress at the seed germination stage, we performed transcriptome sequencing and WGCNA analysis and studied the osmotic regulation and antioxidant response of two quinoa varieties under cold stress conditions. The germination rate of cold-tolerant variety XCq was dramatically higher than that of the cold-sensitive variety QCq, suggesting that they responded differently to cold stress during seed germination (Figure 1A,B). Further comparative transcriptome analysis indicated that the expression patterns of DEGs were significantly different in the two quinoa varieties (Figure 2A–D). Similar results were also found in previous studies in other plant species in response to cold stress [6,11]. Consistent with the distribution trend of DEGs in germinating seeds of waxy corn under cold stress conditions, the number of up-regulated DEGs in XCq was greater than that in QCq [6].

We compared the GO and KEGG enrichment of DEGs identified in the two quinoa varieties (Figure 3A–D). As expected, the enriched GO terms and KEGG pathways in XCq and QCq were significantly different. The up-regulated DEGs were more enriched in GO terms of “seed dormancy process” and “seed maturation” in XCq, but were more enriched in “response to oxidative” and “generation of precursor metabolites and energy” in QCq, implying that the cold stress-tolerant variety may have stronger regulation ability to ensure seed dormancy breaking and germination under cold stress conditions (Figure 3A–B). MAPK cascade is an important signaling module that converts environmental stimuli into cellular responses. It regulates the cold response by modulating the ICE-CBF-COR response pathway during plant response to cold stress [15,32,33]. In *Arabidopsis* and rice, the MAPK cascade pathway interacts with other cold-reactive proteins and TFs and forms a complex regulatory network, such as alteration of the lipid metabolism, accumulation of the osmoprotectant, and induction of the antioxidant defense mechanism, to cope with the cold stress [15,32,34]. Expression of *TaMPK6* from winter wheat enhanced the cold tolerance of transgenic *Arabidopsis* plants through regulating the ICE-CBF-COR module and antioxidant enzyme system [35]. We found that “MAPK signaling pathway-plant” was also significantly enriched in XCq, suggesting that the MAPK cascade pathway could play a key function in seed germination under cold stress conditions (Figure 3C).

Transcription factors function in various biological processes. They are considered as the most important regulators for transcription and signal transduction [28,36,37]. Previously, a total number of 12 key genes related to cold stress, including five AP2/ERF, MYB, NAC, AND C2C2-CO-like transcription factors, were identified in the seedlings of quinoa varieties Dianli281 and Dianli2324 [31]. Similarly, a total number of 32 TFs were also identified in the DEGs in the two quinoa varieties used in our study (Figure 4A). The higher expressions of *bHLH*, *bZIP*, *AP2/ERF*, *HD-ZIP*, *MYB*, *NAC*, *WRKY*, and other *TFs* in XCq compared to QCq after cold stress indicated that large-scale transcriptional changes may have driven its tolerance to cold stress (Figure 4B; Appendix A). The majority of *bHLH*, *bZIP*, *AP2/ERF*, *MYB*, *NAC,* and *WRKY* members were significantly up-regulated, whereas *HD-ZIP* family members were down-regulated, in XCq. Similar to our observation, comparative transcriptomic analyses of seedlings in rice and peanuts with different cold tolerance identified 1583 and 445 TFs, respectively, of which, *bHLH*, *bZIP*, *AP2/ERF*, *MYB/MYB*-related, *NAC*, and *WRKY* families showed to be the most differentially expressed gene members under cold stress conditions [27,28].

As the conserved transcription activator of CBF, ICE1 (inducer of CBF expression 1) in the bHLH family regulated the expression of its downstream genes and enhanced the cold adaptability and resistance of plants [38]. We observed that the expression level of *ICE1* (AUR62027481) in XCq was significantly higher than that in QCq under cold stress conditions, indicating that ICE1 also played an important role in quinoa response to cold stress (Figure 4B). In higher plants, R2R3-MYB TFs regulate the expression of downstream genes to function in low temperature signal transduction. In apple, MdMYB23 activated the expression of *MdCBF1* and *MdCBF2* to enhance the cold tolerance of transgenic plants [39]. We also observed that the expression of three *MYBs* (AUR62033340, AUR62042086, AUR62004765) was significantly up-regulated in XCq upon cold stress treatment, indicating that these *MYBs* also played an important role in quinoa cold tolerance. In addition to R2R3-MYB, bZIP was shown to participate in cold tolerance in rice by interacting with the sugar transport pathway and was also involved in the cold tolerance and sugar transport in the siliques of rapeseed [40,41]. Likewise, bHLHs interacted with COR and ROS to take part in cold tolerance [42]. We found that the expressions of most *bZIPs* and *bHLHs* were significantly higher in XCq than in QCq under cold stress conditions (Figure 4B; Appendix A). In *Arabidopsis*, cold stress up-regulated the expression of MYB15, and the MYB15 protein interacted with ICE1 and bound to the MYB recognition sequence in the promoter sequence of *CBFs* [43]. Compared with normal temperature, the log value of AUR62033340 (*MYBs*) expression under cold stress was 4.6 in XCq, but the value was only 1.00 in QCq. Except for one gene, the log values of up-regulated and down-regulated expressions of most *MYBs* in XCq are higher than those in QCq. The stress response in QCq was stronger, possibly due to its lower tolerance to cold stress than XCq.

RNA-seq and WGCNA has been combined to discover the key genes and interactions that might be functionally related to environmental stress in plants [44]. Through the correlation analysis of gene expression with variety and temperature factors, one module related to variety and two modules related to temperature were selected. For the DEGs in the module related to variety, we found that oxidative stress and redox signaling as well as phytohormone regulation were the most important metabolic pathways. For the DEGs in the modules related to temperature, photosynthesis, oxidative phosphorylation, and steroid biosynthesis were the elements of the metabolic pathways (Figure 5A–C). Therefore, oxidative stress may be an important factor that limits quinoa seed germination under cold stress.

Since plants could alleviate oxidative damage caused by excess ROS via increasing antioxidant enzyme activity and/or antioxidant content, we determined the activities of POD, SOD, and CAT in the germinating seeds [45]. Consistent with the results observed in earlier studies in peach and rapeseed, the activities of SOD and POD were significantly higher in XCq than in QCq under cold stress conditions (Figure 6A–C; [41,46]). Under oxidative stress, the cold-tolerant variety exhibited significantly higher activities of SOD and POD compared to the cold-sensitive variety, indicating its stronger ability to scavenge ROS. MDA accumulation is widely used to evaluate membrane integrity and the stress tolerance of plants. The higher the MDA content, the more serious the plant damage [6]. Upon cold stress treatment, the content of MDA in both XCq and QCq increased, but the content of MDA was significantly lower in XCq than in QCq, indicating that cold stress exacerbated the degree of membrane peroxidation more severely in QCq (Figure 6D). In addition, MDA content was negatively correlated with the activities of SOD, POD, and CAT (Figure 6H). The lower MDA content in XCq could result from the higher antioxidant enzyme activity, which reduced the degree of oxidative damage at the germination stage of quinoa seeds under cold stress conditions. The significant increase in MDA content in the cold-sensitive variety seeds implies that extensive ROS accumulation triggered oxidative damage, which in turn prevented seed germination under low-temperature stress.

MDA values alone were insufficient to reveal the salt tolerance mechanism of quinoa [47]. Additional research into antioxidant enzyme activities and osmoprotectant synthesis is necessary to elucidate the stress tolerance mechanism of quinoa. Previous studies demonstrated that osmotic adjustment played an important role in plant response to abiotic stress [16,48]. Under stress conditions, soluble protein, soluble sugar, and proline could improve the cell water potential, reduce the cytoplasmic freezing point, and protect cellular membranes as an osmotic protective substance in plants [6,46,49]. Quinoa responds to salt and alkali stress by enhancing the synthesis and catabolism of sucrose and starch, thereby increasing the osmotic potential in leaves [50]. The combined analysis of transcriptome and metabolomics highlights the significance of glucose metabolism in quinoa seeds’ response to drought stress [51]. Chilling stress disrupts the balance between the generation and scavenging of ROS, leading to cell death through the destruction of macromolecular structures like proteins, lipids, and DNA [52]. Our results show that the contents of soluble protein and sugar were significantly higher in XCq than in QCq under cold stress conditions (Figure 6A,B,E,F). The higher accumulation of soluble protein and sugar may function in maintaining the stability of the cell membrane structure in XCq to facilitate its tolerance to cold stress. Similar results were also observed in waxy corn, where the contents of soluble sugar and proline in low temperature-resistant waxy corn were significantly higher than that in low temperature-sensitive waxy corn [6]. Furthermore, a significant positive correlation was found between SOD, POD, and CAT activities and soluble protein content (Figure 6H). These results indicate that under oxidative stress, antioxidant enzymes scavenge ROS to protect the mechanisms of protein synthesis within cells, thereby increasing the accumulation of soluble proteins. This synergistic effect helps cells maintain normal physiological functions, enhances their antioxidant capacity, and enables them to better cope with the damage caused by oxidative stress.

## 4. Materials and Methods

### 4.1. Plant Materials and Treatments

Cold-tolerant variety ‘Xingli’ (designated as XCq) and cold-sensitive variety ‘Qingli’ (designated as QCq) were used in this study. Seeds were rinsed in 75% ethanol (*v*/*v*) for 8 min, washed with sterile water three times, and sown on filter paper soaked in 5 mL sterile water in Petri dishes with a diameter of 9 cm. Each Petri dish contained 50 seeds. For cold stress treatment, seeds were germinated at 4  °C, while for the control group, seeds were germinated at 20 °C. The samples were cultured in an artificial climate chamber with an illumination condition of 0 lx and a relative humidity (RH) of 70%.Seed germination was recorded on a daily basis. Seeds with radicle length exceeding 2 mm were considered germinated.

### 4.2. RNA-Seq Analysis

Total RNA was isolated from seed embryos at the germination stages of 36 h (20 °C) and 76 h (4  °C) using a FastPure Universal Plant Total RNA Isolation Kit (Vazyme, Nanjing, China). The quality of total RNA was measured using a micro-spectrophotometer (OD260/280). The prepared libraries were sequenced on the Illumina Novaseq6000 platform (Majorbio, Shanghai, China), and 250 bp paired-end reads were generated. Clean data were obtained by filtering the raw data and then mapping to the quinoa reference genome [version *Chenopodium quinoa* v1.0] (https://phytozome-next.jgi.doe.gov/info/Cquinoa_v1_0 (accessed on 27 March 2024)) using HISAT2.2.4 [53]. Gene transcript abundance quantification and abundance difference examination were conducted as described by Love [54]. For the analysis of differentially expressed genes (DEGs), the DESeq2 R package was used. Genes with an adjusted *p*-value < 0.05 (*p.adj  *<  0.05) and the difference multiplier satisfying |log_2_(FoldChange)| ≥ 2, were thought to be differentially expressed. Gene ontology (GO) enrichment and Kyoto Encyclopedia of Gene and Genome (KEGG) pathway analysis were performed using the clusterProfiler package. GO terms and KEGG pathways with an adjusted *p*-value (*p.adj*) < 0.05 were thought to be significantly enriched.

### 4.3. Identification of Co-Expression Modules

The weighted gene co-expression network analysis (WGCNA) package in R software (Version 3.4.4) was used to identify modules with high correlation to low temperature response in quinoa. The genes with similar expression patterns were clustered into gene modules through average linkage hierarchical clustering, using the TOM-based phase dissimilarity metric with *minModuleSize* = 50, *MEDissThres* = 0.25, and *power * =  10 [55]. The modules with the strongest relevance to temperature and variety were selected as the key modules for subsequent analysis.

### 4.4. Physiological Indicator Analysis

Seed embryos at germination stages of 36 h (20 °C) and 76 h (4  °C) were used for physiological indicator analysis. The activities of SOD, POD, and CAT and the content of MDA were detected as depicted by Guo [42]. Soluble protein was determined using the Coomassie brilliant blue method, as described previously [56]. Soluble sugar and proline were quantified with commercial kits (Solarbio, Shanghai, China).

### 4.5. Validation of RNA-Seq Data

High-quality RNAs extracted from seed embryos at the germination stages of 36 h (20 °C) and 76 h (4  °C) were reversely transcribed into cDNA for qRT-PCR. Reverse transcription of total RNA was conducted with the HiScript II 1st Strand cDNA Synthesiskit (Vazyme Biotech, Shanghai, China). qRT-PCR programs were performed on a QuantStudio 3 Real-Time PCR System ((Thermo Fisher Scientific Inc., Waltham, MA, USA) using ChamQ^®^SYBR qPCR Master Mix (Vazyme Biotech, Shanghai, China). The 2^–ΔΔCt^ method was employed for normalizing the relative expression of each gene using *ELF1a* as an internal reference [57]. For each qRT-PCR experiment, three independent biological replicates were conducted. Primer sequences are given in Appendix A.

### 4.6. Statistical Analysis

Student’s *t*-test was performed with the Rbase package. Mean values and standard deviations (SDs) were calculated from three biological and three technical replicates. Significant differences at *p* < 0.05 were determined with the multiple range test.

## 5. Conclusions

Taken together, comparative transcriptomic analysis between cold-tolerant variety XCq and cold-sensitive variety QCq in response to cold stress was performed. A total number of 4552 and 4845 DEGs were identified, respectively. Compared with QCq, DEGs involved in the MAPK signaling and oxidative phosphorylation metabolic pathways were significantly enriched in XCq. TF families, such as bHLH, ERF/DREB, and MYB, involved in the ICE-CBF-COR cascade, as well as bZIP, NAC, and WRKY, and soluble sugar and antioxidant enzymes, were closely associated with the cold tolerance of quinoa during seed germination. Upon cold stress, the expression of genes involved in the ICE-CBF-COR cascade and oxidative phosphorylation metabolic pathway was activated, leading to improved tolerance to cold stress (Figure 8).

## Figures and Tables

**Figure 1 plants-14-01212-f001:**
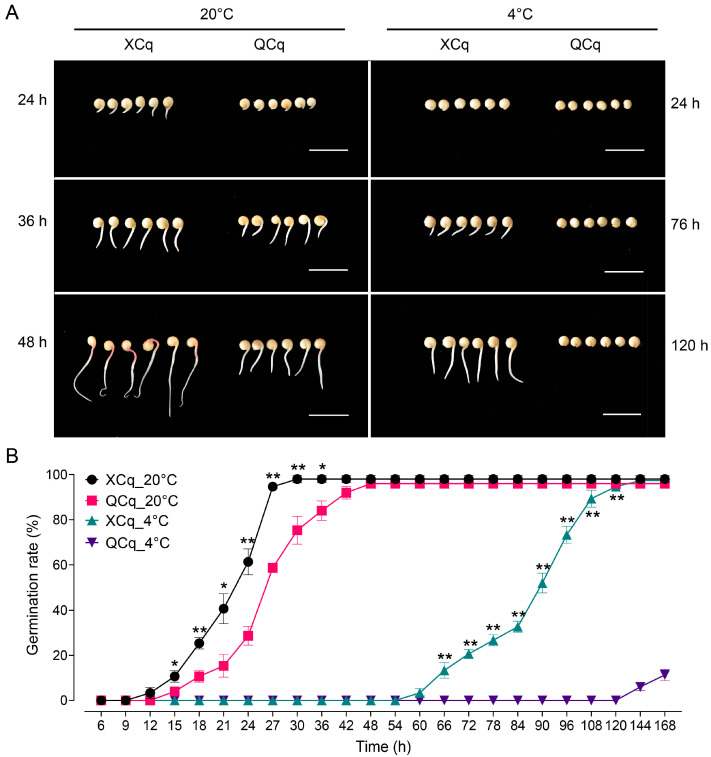
Germination assays. Seeds of XCq and QCq were germinated for different time periods undernormal (20 °C) and cold stress (4 °C) conditions. (**A**) Images showing the germination of XCq and QCq seeds. (**B**) Germination rates of XCq and QCq seeds. Values are means ± standard deviation (SD) from three biological replicates (*n* = 3). “*” and “**” represent significant difference between XCq and QCq at *p* < 0.05 and 0.01, respectively.

**Figure 2 plants-14-01212-f002:**
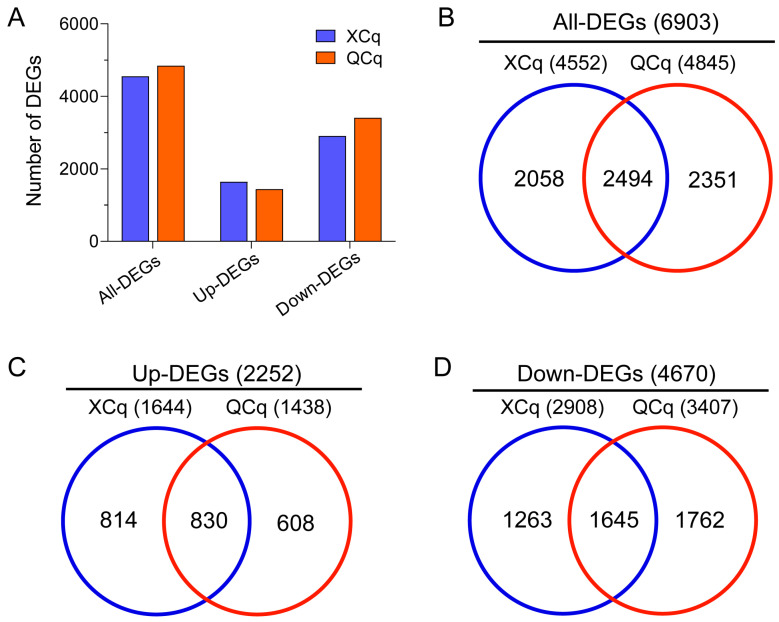
Differentially expressed genes (DEGs) between the seed embryos germinated at 20 °C and 4 °C in XCq and QCq. (**A**) A summary of the numbers of total (All), up-regulated (Up), and down-regulated (Down) DEGs. (**B**–**D**) Venn diagrams to show the numbers of all, up- and down-regulated DEGs in XCq and QCq.

**Figure 3 plants-14-01212-f003:**
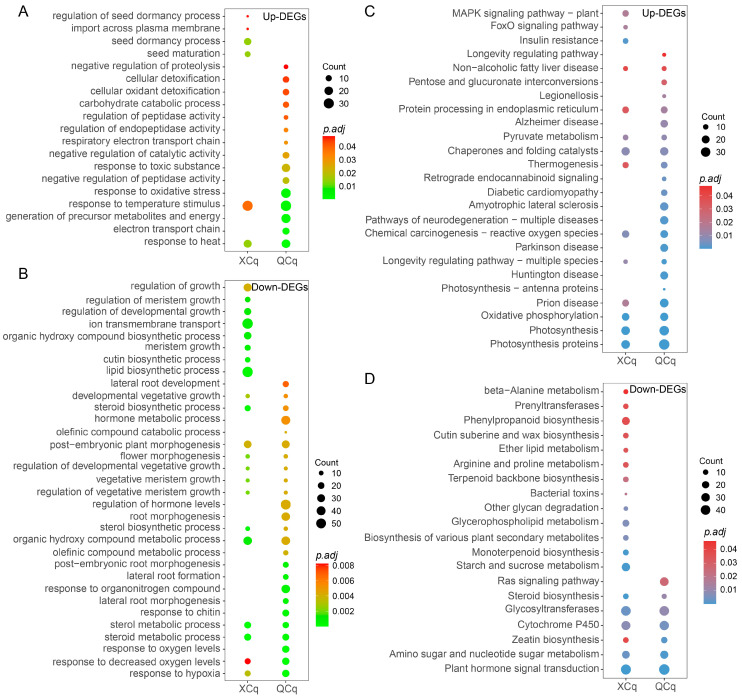
Gene ontology-biological process (GO-BP) and the Kyoto Encyclopedia of Genes and Genomes (KEGG) pathway enrichment analysis of the DEGs for XCq and QCq. (**A**) Scatter plots showing the up-regulated GO-BP categories in XCq and QCq. (**B**) Scatter plots showing the down-regulated GO-BP categories in XCq and QCq. (**C**) Scatter plots showing the up-regulated KEGG categories in XCq and QCq. (**D**) Scatter plots showing the down-regulated KEGG categories in XCq and QCq. The rich factor represents the ratio of the number of DEGs enriched by the pathway to the number of annotated genes. The color bar represents the significance test *p*-value adjusted for multiple hypothesis testing.

**Figure 4 plants-14-01212-f004:**
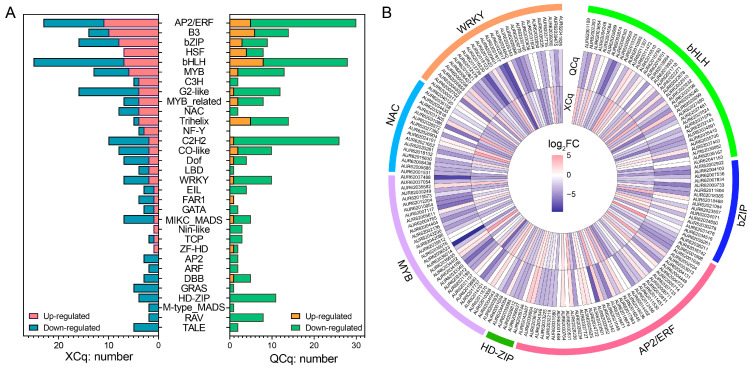
Transcription factor (TF) genes among the identified DEGs. (**A**) Up- and down-regulated TF genes in XCq and QCq in responsive to cold stress. (**B**) Heatmap showing the expression levels of the top 7 TF family genes. The differential transcript levels calculated as log_2_(4 °C/20 °C) are shown in the color legend. The color red and blue indicates up- and down-regulated genes, respectively.

**Figure 5 plants-14-01212-f005:**
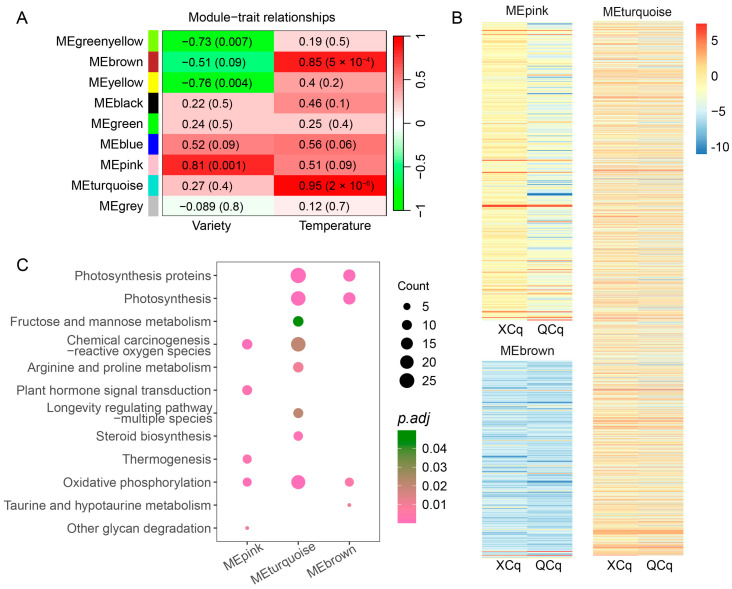
Weighted gene co-expression network analysis (WGCNA) of the DEGs identified in XCq and QCq. (**A**) Correlation of the co-expression modules with the quinoa variety and temperature. Red and green colors represent the positive and negative correlation with gene expression, respectively. (**B**) Heatmaps displaying the expression levels of the DEGs clustered in the pink and turquoise modules. (**C**) Scatter plots showing the results of the KEGG pathway enrichment analysis of the DEGs clustered in the pink and turquoise modules.

**Figure 6 plants-14-01212-f006:**
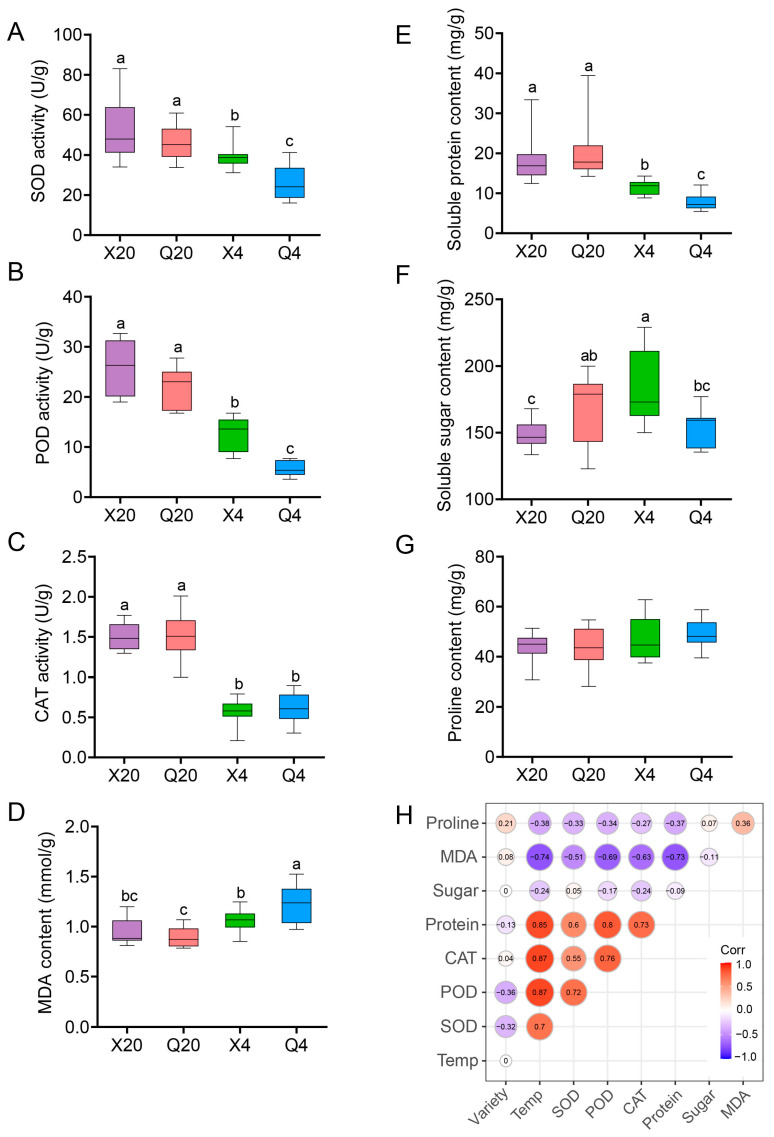
Antioxidative stress and Pearson correlation analysis. Seed embryos of XCq and QCq at the germination stages of 36 h (20 °C) and 76 h (4  °C) were used. (**A**–**C**) Activities of superoxide dismutase (SOD), peroxide dismutase (POD), and catalase (CAT). (**D**–**G**) Contents of malondialdehyde (MDA), soluble protein, soluble sugar, and proline. (**H**) Correlation of antioxidant enzyme activities and osmoprotectant contents with quinoa variety and temperature. X20, XCq germinated at 20 °C; Q20, QCq germinated at 20 °C; X4, XCq germinated at 4 °C; Q4, QCq germinated at 4 °C. Different letters represent significant difference at *p* < 0.05.

**Figure 7 plants-14-01212-f007:**
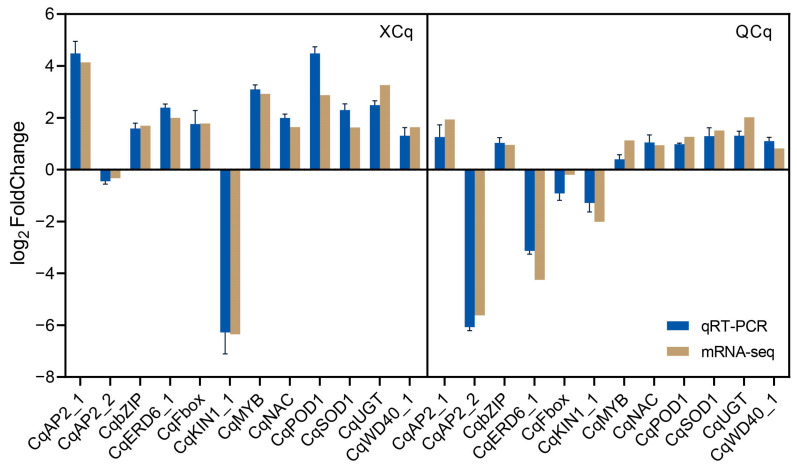
Verification of RNA-seq data with quantitative real-time PCR. Column in khaki represents fold changes based on FPKM calculated from globally normalized RNA-seq data. Column in blue with standard errors indicates fold changes based on the relative expression level determined with qRT-PCR using the 2^−ΔΔCT^ method. Three biological replicates under normal temperature (20 °C) and cold stress (4 °C) conditions were conducted.

**Figure 8 plants-14-01212-f008:**
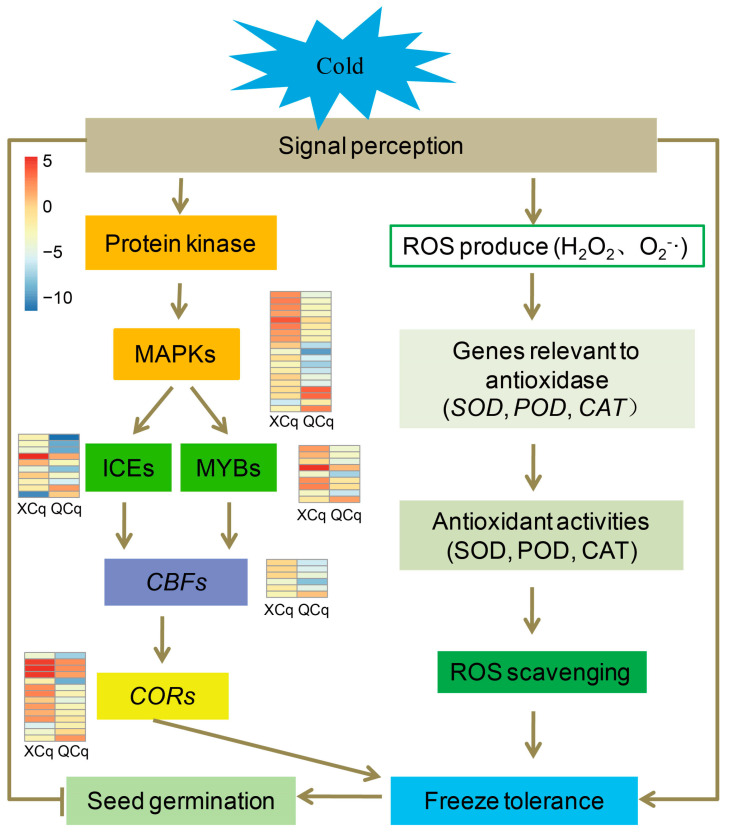
Proposed working model for the cold tolerance of quinoa during seed germination. Heatmaps indicate the expression profilings of DEGs related to the MAPK-mediated ICE-CBF-COR cascade and antioxidant pathway.

## Data Availability

All raw reads were deposited in the Sequence Read Archive (SRA) database in NCBI, with accession number PRJNA1034514.

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
