# Peer review of "Comparative Transcriptomic Analyses Reveal Key Pathways in Response to Cold Stress at the Germination Stage of Quinoa (*Chenopodium quinoa* Willd.) Seeds"

_plants, 2025, doi:10.3390/plants14081212_

Round 1

Reviewer 1 Report

Comments and Suggestions for Authors

Generally, the paper is well organized and structured, but I notice some awkward phrasing; the article will benefit from a good language editor. Examples are:

  • Line 21: “world widely” Versus worldwide or widely??
  • Line 34 and 61: “Peroridase” versus Peroxidase
  • Line 244: Closed Correlated Versus Closely Correlated.
  • Line 85: remove “t” I think that is typo.

A brief introduction to some uncommon technical methods like WGCNA R will be helpful

Comments on the Quality of English Language

The English language is good, but it appears the authors' first language is not English. The article will benefit from a thorough review by an editor with proficiency in English.

Author Response

Point by point response to Reviewer 1

  1. Comments and Suggestions for Authors

Generally, the paper is well organized and structured, but I notice some awkward phrasing; the article will benefit from a good language editor. Examples are:

  • Line 21: “world widely” Versus worldwide or widely??
  • Line 34 and 61: “Peroridase” versus Peroxidase
  • Line 244: Closed Correlated Versus Closely Correlated.
  • Line 85: remove “t” I think that is typo.

Response1: Thank you for your comments. We have carefully revised the manuscript thoroughly.

(1) “world widely” has been modified to “widely”. (Line 21)

(2) “Peroridase” has been modified to “Peroxidase”. (Line 34 and 61)

(3) “Closed Correlated” has been modified to “Closely Correlated”. (Line 244)

(4) Thank you. “t” has been modified to “those”. (Line 244)

A brief introduction to some uncommon technical methods like WGCNA R will be helpful.

Response1: A brief introduction about the usage of WGCNA has been added in the "4.3. Identification of Co-Expression Modules" section.

  1. Comments on the Quality of English Language

The English language is good, but it appears the authors' first language is not English. The article will benefit from a thorough review by an editor with proficiency in English.

Response2: Thank you for your comments. We have asked Dr. Jessie Zhang at University of Toronto, Canada, to have a thorough review for the revised version of this manuscript.

Reviewer 2 Report

Comments and Suggestions for Authors

Comment 01

The manuscript titled "Comparative Transcriptomic Analyses Reveal Key Pathways in Response to Cold Stress during the Germination of Quinoa (Chenopodium quinoa Willd.) Seeds" (ID: plants-3562071) presents a comprehensive transcriptomic analysis aimed at understanding the molecular mechanisms underlying cold stress responses in quinoa seeds, focusing on the germination stage. The study compares the cold-tolerant cultivar XCq with the cold-sensitive cultivar QCq, identifying 4552 and 4845 differentially expressed genes (DEGs), respectively, under cold stress. The Kyoto Encyclopediaa of Genes and Genomes (KEGG) pathway analysis revealed up-regulation of key genes involved in cold response, such as those in the MAPK signaling pathway and various transcription factors, specifically in XCq. The study further highlights the role of oxidative phosphorylation and oxidative metabolism pathways in cold tolerance, supported by physiological data showing higher enzyme activities and metabolite levels in the cold-tolerant cultivar. This research provides valuable insights into the genetic basis of cold tolerance in quinoa, offering potential targets for molecular breeding efforts aimed at improving crop resilience to cold stress. The findings have significant implications for enhancing the productivity and sustainability of quinoa cultivation in colder climates, with potential socio-economic benefits for global food security, especially in regions where quinoa is an emmerging crop.

Comment 02

The manuscript exhibits a high percentage of textual overlap, with an iThenticate report indicating a 36% match. While it is common for certain phrases and references to overlap with previous literature, this percentage is relatively high and warants attention.

Comment 03

The experimental work presented in the manuscript appears to be well-conducted and thoroughly planned. The authors have employed a robust comparative transcriptomic analysis to investigate the cold stress response in quinoa seeds, comparing a cold-tolerant and a cold-sensitive cultivar. The integration of various analytical methods, such as KEGG pathway analysis and Weighted Gene Co-Expression Network Analysis (WGCNA), demonstrates a comprehensive approach to understanding the molecular mechanisms involved.

Comment 04

The manuscript is well-written and clearly organised.

Comment 05

The Graphical representations are of excellent quality and appear highly professional. They effectively complement the data presented and enhance the clarity of the results. However, one issue that needs attention is the small font size used in some of the figures, which makes it difficult to read the text in certain instances. It is essential to address this issue by increasing the font size to ensure that all textual elements are legible and easily comprehensible to readers.

Comment 06:

 Experimental Design Clarification: While the experimental design appears robust, particularly in comparing the two quinoa cultivars under cold stress, could the authors elaborate on why the specific time points (36 h a nd 76 h) were chosen for the transcriptomic and physiochemical analyses? was there any specific rationale behind selecting these hours, and do they correspond to critical stages of cold stress s response ?

Comment 07:

Data Availability and Validation: The authors mention the use  of RNA-seq data, but the methods for validating the findings through qRT-PCR are su mmarised briefly. Could the authors provide more detail on how the qRT- PCR data correlate with the RNA -seq data, especially for genes associated with cold tolerance?  This would help strengthen the reliaBbility of the findings.

Comment 08

The manuscript includes only a single recent reference from 2024, which may not sufficiently reflect the latest developments in the field. i recommend expanding the list of recent references to include additional studies published in the past year or two, particularly those that are directly relevant to the topics of  cold stress, transcriptomic analyses, and quinoa research.

Comment 09

Please change "during the Germination of" to "at the Germination Stage of".

Please change "were respectively identified" to "were identified, respectively".

Please change "would be" to "can be".

Please change "stress condition" to "stress conditions".

Please change "were respectively germinated" to "were germinated, respectively".

Please change "cold stress tolerant variety" to "cold stress-tolerant variety".

Please change "was tested with" to "was measured using".

Please change "the functions of the potential candidate genes" to "the functions of potential candidate genes".

Comment 10

  • Please, make sure that all references have a corresponding citation within the text and vice versa.
  • Please, double-check the spelling of the author’s names and dates and make sure they are correct and consistent with the citations.
  • Please, spell out all journal titles in the references section.
  • Please, make sure that all figures and tables are cited within the text and that they are cited in consecutive order.
  • Please, spell al the abbreviations the first timme when they are mentioned in the text.

Author Response

Point by point response to Reviewer 2

Comments and Suggestions for Authors

Comment 01

The manuscript titled "Comparative Transcriptomic Analyses Reveal Key Pathways in Response to Cold Stress during the Germination of Quinoa (Chenopodium quinoa Willd.) Seeds" (ID: plants-3562071) presents a comprehensive transcriptomic analysis aimed at understanding the molecular mechanisms underlying cold stress responses in quinoa seeds, focusing on the germination stage. The study compares the cold-tolerant cultivar XCq with the cold-sensitive cultivar QCq, identifying 4552 and 4845 differentially expressed genes (DEGs), respectively, under cold stress. The Kyoto Encyclopediaa of Genes and Genomes (KEGG) pathway analysis revealed up-regulation of key genes involved in cold response, such as those in the MAPK signaling pathway and various transcription factors, specifically in XCq. The study further highlights the role of oxidative phosphorylation and oxidative metabolism pathways in cold tolerance, supported by physiological data showing higher enzyme activities and metabolite levels in the cold-tolerant cultivar. This research provides valuable insights into the genetic basis of cold tolerance in quinoa, offering potential targets for molecular breeding efforts aimed at improving crop resilience to cold stress. The findings have significant implications for enhancing the productivity and sustainability of quinoa cultivation in colder climates, with potential socio-economic benefits for global food security, especially in regions where quinoa is an emmerging crop.

Response1: Thank you for your comments.

Comment 02

The manuscript exhibits a high percentage of textual overlap, with an iThenticate report indicating a 36% match. While it is common for certain phrases and references to overlap with previous literature, this percentage is relatively high and warants attention.

Response2: Thank you. Following you suggestion, we have revised the manuscript thoroughly and decreased the percentage of textual overlap.

Comment 03

The experimental work presented in the manuscript appears to be well-conducted and thoroughly planned. The authors have employed a robust comparative transcriptomic analysis to investigate the cold stress response in quinoa seeds, comparing a cold-tolerant and a cold-sensitive cultivar. The integration of various analytical methods, such as KEGG pathway analysis and Weighted Gene Co-Expression Network Analysis (WGCNA), demonstrates a comprehensive approach to understanding the molecular mechanisms involved.

Response: Thank you for your comments.

Comment 04

The manuscript is well-written and clearly organised.

Response4: Thank you for your comments.

Comment 05

The Graphical representations are of excellent quality and appear highly professional. They effectively complement the data presented and enhance the clarity of the results. However, one issue that needs attention is the small font size used in some of the figures, which makes it difficult to read the text in certain instances. It is essential to address this issue by increasing the font size to ensure that all textual elements are legible and easily comprehensible to readers.

Response5: Thank you. We have increased the font size in Figure 3, 4 and 6, to ensure that all textual elements are legible and easily comprehensible to readers.

Comment 06

Experimental Design Clarification: While the experimental design appears robust, particularly in comparing the two quinoa cultivars under cold stress, could the authors elaborate on why the specific time points (36 h and 76 h) were chosen for the transcriptomic and physiochemical analyses? was there any specific rationale behind selecting these hours, and do they correspond to critical stages of cold stress s response ?

Response6: Yes, they correspond to critical stages of cold stress response. Under normal condition, the seeds of both XCq and QCq exhibited good germination at 36 h, with a radicle length exceeding 2 mm. In contrast, under cold stress condition, neither XCq nor QCq seeds germinated at 36 h. Therefore, no sampling was conducted at this time point. Under cold stress condition, the seeds of QCq failed to germinate throughout the entire cold stress treatment period, whereas the seeds of XCq germinated well at 76 h,with a radicle length exceeding 2 mm. Based on these observations, we collected samples at this time point for further analysis.

Comment 07

Data Availability and Validation: The authors mention the use of RNA-seq data, but the methods for validating the findings through qRT-PCR are su mmarised briefly. Could the authors provide more detail on how the qRT- PCR data correlate with the RNA -seq data, especially for genes associated with cold tolerance? This would help strengthen the reliaBbility of the findings.

Response7: qRT-PCR analysis was carried out to confirm the reliability of RNA-seq data. Due to the large number of differentially expressed genes (DEGs), it was not feasible to validate the expression levels of all DEGs using qRT-PCR. Based on the RNA-seq results, 12 cold tolerance associated genes with substantial expression differences between XCq and QCq were selected for validation.

Comment 08

The manuscript includes only a single recent reference from 2024, which may not sufficiently reflect the latest developments in the field. i recommend expanding the list of recent references to include additional studies published in the past year or two, particularly those that are directly relevant to the topics of cold stress, transcriptomic analyses, and quinoa research.

Response8: Drought and salinity are the abiotic stresses mostly studied in quinoa; however, studies of other important stress factors, such as heat, cold, heavy metals, and UV-B light irradiance, are limited. We added three recent references relevant to abiotic stress response of quinoa. Related contents have been added in the "3. Discussion" section.

[45] Serrat, X.; Quello, A.; Manikan, B.; Lino, G.; Nogués, S. Comparative salt-stress responses in salt-tolerant (Vikinga) and salt-sensitive (Regalona) quinoa varieties. Physiological, anatomical and biochemical perspectives. Agronomy 2024, 14, 3003.

[50] Bao, Q.; Wu, Y.; Wang, Y.; Zhang, Y. Comparative Transcriptomic analysis reveals transcriptional differences in the response of quinoa to salt and alkali stress responses. Agronomy 2024, 14, 1596.

[51] Wang, C.; Lu, C.; Wang, J.; Liu, X.; Wei, Z.; Qin, Y.; Zhang, H.; Wang, X.; Wei, B.; Lv, W.; Mu, G. Molecular mechanisms regulating glucose metabolism in quinoa (Chenopodium quinoa Willd.) seeds under drought stress. BMC Plant Biol. 2024, 24, 796.

Comment 09

Please change "during the Germination of" to "at the Germination Stage of".

Response9: Thank you. We have corrected three such mistakes throughout the manuscript.

Please change "were respectively identified" to "were identified, respectively".

Response9: Thank you. We have corrected two such mistakes throughout the manuscript.

Please change "would be" to "can be".

Response9: Thank you. We have corrected this mistake in the manuscript.

Please change "stress condition" to "stress conditions".

Response: Thank you. We have corrected sixteen such mistakes throughout the manuscript.

Please change "were respectively germinated" to "were germinated, respectively".

Response9: Thank you. We have corrected two such mistakes throughout the manuscript.

Please change "cold stress tolerant variety" to "cold stress-tolerant variety".

Response9: Thank you. We have corrected this mistake in the manuscript.

Please change "was tested with" to "was measured using".

Response9: Thank you. We have corrected this mistake in the manuscript.

Please change "the functions of the potential candidate genes" to "the functions of potential candidate genes".

Response9: Thank you. We have corrected this mistake in the manuscript.

Comment 10

  • Please, make sure that all references have a corresponding citation within the text and vice versa.

Response10: Thank you. All references have been checked, and they have corresponding citations within the text and that all citations are properly referenced.

  • Please, double-check the spelling of the author’s names and dates and make sure they are correct and consistent with the citations.

Response10: Thank you. The spelling of the authors’ names and the dates has been checked to ensure they are correct and consistent with the citations.

  • Please, spell out all journal titles in the references section.

Response10: Thank you. All journal titles in the references section have been spelled out.

  • Please, make sure that all figures and tables are cited within the text and that they are cited in consecutive order.

Response10: Thank you. All figures and tables have been checked, and they are cited in consecutive order.

  • Please, spell al the abbreviations the first time when they are mentioned in the text.

Response10: Thank you. All abbreviations have been spelled when they are mentioned first time in the text. e.g. reactive oxygen species (ROS), differentially expressed genes (DEGs) and weighted gene co-expression network analysis (WGCNA).

Reviewer 3 Report

Comments and Suggestions for Authors

Comments

- The authors state that the molecular mechanism of the response to cold stress during seed germination is still largely unknown. However, considering the regeneration status of pseudocereals in cold environments, they should express their opinions on the mechanism of cold stress.

- Have the gene transfer processes of XCq and QCq cold tolerant and sensitive quinoa varieties been tested in the locations where both varieties are grown and how the changes in the bioactive properties of the seeds occur as a result of the effect of ecological conditions have been observed? Because, as a result of gene transfer, the active substance properties of hybrid varieties can vary significantly compared to individual varieties.

- What are the factors that indicate that DEGs encoding transcription factors in the mitogen-activated protein kinase (MAPK) signaling pathway and AP2/ERF, bHLH, bZIP, MYB, ICEs, and CORs related to the cold response are uniquely upregulated in XCq, and the mechanism of action of these factors needs to be elucidated.

- It should be specified whether the factor that causes the significant enrichment of DEGs clustered in co-expression modules positively associated with the factors of quinoa variety and temperature in the oxidative phosphorylation metabolic pathway is due to the temperature during the day or the narrowing of the temperature gap between day and night. Because, depending on the location, the difference between day and night is an effective factor in both vegetative and generative development in plants.

- The mechanism by which antioxidant enzymes increase soluble protein and sugar in the event of oxidative stress should be explained.

- It should be clarified whether the stress in cold-sensitive quinoa varieties is due to the cold or to the free radicals they accumulate in large amounts in their bodies.

Comments on the Quality of English Language

suitable

Author Response

Point by point response to Reviewer 3

Comments and Suggestions for Authors

Comments

- The authors state that the molecular mechanism of the response to cold stress during seed germination is still largely unknown. However, considering the regeneration status of pseudocereals in cold environments, they should express their opinions on the mechanism of cold stress.

Response1: Thank you for your comments. The Description of relevant mechanisms have been added to the "Abstract" section, as follows: “Our findings revealed that the enhanced activities of antioxidant enzymes alleviate lipid peroxidation of membranes and promote the accumulation of osmotic adjustment substances, thereby enabling seeds to better resist oxidative damage under cold stress.”

- Have the gene transfer processes of XCq and QCq cold tolerant and sensitive quinoa varieties been tested in the locations where both varieties are grown and how the changes in the bioactive properties of the seeds occur as a result of the effect of ecological conditions have been observed? Because, as a result of gene transfer, the active substance properties of hybrid varieties can vary significantly compared to individual varieties.

Response2: Quinoa is a self-pollinating plant. To minimize the risk of pollen spread, we employed physical isolation techniques by propagating the seeds in two separate experimental fields located at a considerable distance from each other during the breeding process. Throughout the planting period, we regularly monitored the growth of the plants and promptly removed any individuals whose traits were significantly different from those of the surrounding plants, as these may have been impacted by cross-pollination.

- What are the factors that indicate that DEGs encoding transcription factors in the mitogen-activated protein kinase (MAPK) signaling pathway and AP2/ERF, bHLH, bZIP, MYB, ICEs, and CORs related to the cold response are uniquely upregulated in XCq, and the mechanism of action of these factors needs to be elucidated.

Response3: We apologize that the statement in the text was incorrect. The necessary revisions have been made in "Abstract" section, as follows: Originally stated as, “Kyoto Encyclopedia of Genes and Genomes (KEGG) pathway analysis demonstrated that DEGs involved in the mitogen activated protein kinase (MAPK) signaling pathway and encoding transcription factors, such as AP2/ERF, bHLH, bZIP, MYB, ICEs and CORs related to cold response, were uniquely up-regulated in XCq.” It has now been modified to: “Kyoto Encyclopedia of Genes and Genomes (KEGG) pathway analysis demonstrated that the mitogen activated protein kinase (MAPK) signaling pathway was identified only among the up-regulated DEGs in XCq. The expression of DEGs, which encoding transcription factors, such as AP2/ERF, bHLH, bZIP, MYB, ICEs and CORs related to cold response, were higher in XCq than in QCq in response to cold stress.”

- It should be specified whether the factor that causes the significant enrichment of DEGs clustered in co-expression modules positively associated with the factors of quinoa variety and temperature in the oxidative phosphorylation metabolic pathway is due to the temperature during the day or the narrowing of the temperature gap between day and night. Because, depending on the location, the difference between day and night is an effective factor in both vegetative and generative development in plants.

Response4: We apologize for the lack of clarity in our description of the methods. The germination experiment for quinoa seeds was carried out in complete darkness with a constant temperature. Specifically, for the cold stress treatment, seeds were germinated at 4°C, whereas for the control group, seeds were germinated at 20°C. All samples were cultured in an artificial climate chamber under conditions of 0 lx illumination and 70% relative humidity (RH). This information has now been incorporated into the "4.1. Plant Materials and Treatments" section.

- The mechanism by which antioxidant enzymes increase soluble protein and sugar in the event of oxidative stress should be explained.

Response5: The explanations regarding to the mechanisms have been added to the "3. Discussion" section, as follows: “ Furthermore, a positive correlation was found between SOD, POD and CAT activities and soluble protein content (Figure 6H). These results indicate that under oxidative stress, antioxidant enzymes scavenge ROS to protect the mechanisms of protein synthesis within cells, thereby increasing the accumulation of soluble proteins. This synergistic effect helps cells maintain normal physiological functions, enhances their antioxidant capacity, and enables them to better cope with the damage caused by oxidative stress.”

- It should be clarified whether the stress in cold-sensitive quinoa varieties is due to the cold or to the free radicals they accumulate in large amounts in their bodies.

Response6: A description has been added to the "3. Discussion" section, as follows:

“Under oxidative stress, the cold-tolerant variety exhibited significantly higher activities of SOD and POD compared to the cold-sensitive variety, indicating its stronger ability to scavenge ROS.”

“The lower MDA content in XCq could be resulted from the higher antioxidant enzyme activities, which reduced the degree of oxidative damage at the germination stage of quinoa seeds under cold stress conditions. The significant increase in MDA content in cold sensitive variety seeds implies that extensive ROS accumulation has triggered oxidative damage, which in turn prevents seed germination under low temperature stress.”
